

# Microbial biomarker detection in shrimp larvae rearing water as putative bio-surveillance proxies in shrimp aquaculture

Nolwenn Callac[1], Carolane Giraud[1,2], Viviane Boulo[1,3], Nelly Wabete[1] and Dominique Pham[1]

[1] Ifremer, IRD, Université de la Nouvelle-Calédonie, Université de La Réunion, CNRS, UMR 9220 ENTROPIE, Ifremer, Nouméa, New-Caledonia
[2] Institut des Sciences Exactes et Appliquées, University of New Caledonia, Nouméa, New-Calédonia
[3] IHPE, Université de Montpellier, CNRS, Ifremer, Université de Perpignan via Domitia, Ifremer, Montpellier, France

Corresponding author
Nolwenn Callac,
Nolwenn.Callac@ifremer.fr

## ABSTRACT

**Background**. Aquacultured animals are reared in water hosting various microorganisms with which they are in close relationships during their whole lifecycle as some of these microorganisms can be involved in their host's health or physiology. In aquaculture hatcheries, understanding the interactions existing between the natural seawater microbiota, the rearing water microbiota, the larval stage and the larval health status, may allow the establishment of microbial proxies to monitor the rearing ecosystems. Indeed, these proxies could help to define the optimal microbiota for shrimp larval development and could ultimately help microbial management.

**Methods**. In this context, we monitored the daily composition of the active microbiota of the rearing water in a hatchery of the Pacific blue shrimp *Penaeus stylirostris*. Two distinct rearing conditions were analyzed; one with antibiotics added to the rearing water and one without antibiotics. During this rearing, healthy larvae with a high survival rate and unhealthy larvae with a high mortality rate were observed. Using HiSeq sequencing of the V4 region of the 16S rRNA gene of the water microbiota, coupled with zootechnical and statistical analysis, we aimed to distinguish the microbial taxa related to high mortality rates at a given larval stage.

**Results**. We highlight that the active microbiota of the rearing water is highly dynamic whatever the larval survival rate. A clear distinction of the microbial composition is shown between the water harboring heathy larvae reared with antibiotics *versus* the unhealthy larvae reared without antibiotics. However, it is hard to untangle the effects of the antibiotic addition and of the larval death on the active microbiota of the rearing water. Various active taxa of the rearing water are specific to a given larval stage and survival rate except for the zoea with a good survival rate. Comparing these communities to those of the lagoon, it appears that many taxa were originally detected in the natural seawater. This highlights the great importance of the microbial composition of the lagoon on the rearing water microbiota. Considering the larval stage and larval survival we highlight that several genera: *Nautella, Leisingera, Ruegerira, Alconivorax, Marinobacter* and *Tenacibaculum,* could be beneficial for the larval survival and may, in the rearing water, overcome the r-strategist microorganisms and/or putative pathogens. Members of these genera might also act as probiotics for the larvae.

*Marivita*, *Aestuariicocccus*, HIMB11 and *Nioella*, appeared to be unfavorable for the larval survival and could be associated with upcoming and occurring larval mortalities. All these specific biomarkers of healthy or unhealthy larvae, could be used as early routine detection proxies in the natural seawater and then during the first days of larval rearing, and might help to manage the rearing water microbiota and to select beneficial microorganisms for the larvae.

## INTRODUCTION

In New-Caledonia, hatcheries of the Pacific blue shrimp *Penaeus stylirostris* face high larval mortality rates (*Beliaeff et al., 2009*; *Pham et al., 2012*). The causes of such mortalities are not yet understood and multi-factorial reasons seem to trigger larval death. For example, only 128 million post larvae were produced in 2019; while the production had reached up to 167 million post larvae in 2005 (https://www.isee.nc/economie-entreprises/entreprises-secteurs-d-activites/agriculture-peche-aquaculture, section Les structures aquacoles et maritimes). These larval mortalities create many issues, as not enough post larvae are available to be spread among the earthen ponds of the 18 farms of the territory, where they grow until they reach a certain weight to be sold. This induces a commercial deficit and an economical loss for the farmers, the workers and the territory. Several factors, such as water quality or bacterial infections, could play a role in the larval mortalities but these hypotheses have been denied. Indeed, no known pathogen was found, and various larval survival rates can be observed among larvae reared under the same conditions in the same rearing water. However, we think that dysbiosis of the rearing water microbiota could hamper the larval survival. Indeed, aquacultured animals are reared in water hosting various microorganisms with which they are in close relationships during their whole lifecycle as some of these microorganisms can be involved in their host's health, physiology and fitness (*Goarant et al., 2006*; *Ganguly & Prasad, 2012*; *Carbone & Faggio, 2016*; *Zheng et al., 2017*; *Sun et al., 2019*; *Wei et al., 2020*; *Wang et al., 2020b*; *Angthong et al., 2020*). It has been shown that the rearing water microbiota could influence pre-feeding fish larvae (*Bledsoe et al., 2016*; *Wilkes Walburn et al., 2019*) and may contribute to larval health in cod larvae (*Lauzon et al., 2010*). Regarding shrimps, it has been proved that the rearing water microbiota can interact with the shrimps and that some microbial taxa originating from the water can be transmitted to the host microbiota at various lifecycle stages (*Huang et al., 2018*; *Giraud et al., 2021*). Thus, in order to establish microbial proxies to monitor the rearing ecosystems in shrimp hatcheries, it seems necessary to uncover the interactions existing between the natural seawater microbiota, the rearing water microbiota, the larval stage and the larval health status. To date, a few studies have been conducted on both diseased and healthy *Penaeus vannamei* shrimps at various lifecycle stages. In these studies, the authors managed to establish links between the microbial communities and the health status of the

shrimp, and also identified bacterial indicators of diseased shrimps (*Xiong, Zhu & Zhang, 2014*; *Zheng et al., 2017*). However rearing methods of *P. vannamei* and *P. stylirostris* larvae slightly differ. Indeed, *P. vannamei* larvae are reared in oceanic water using both probiotics and antibiotics while, in New-Caledonia, larval rearing is performed using lagoon seawater and antibiotics are often only added until post-larval stage (*Pham et al., 2012*; *Zheng et al., 2017*). Thus, in order to overcome the larval deficit in New-Caledonia, it is imperative to establish microbial proxies to monitor the rearing ecosystems and to distinguish taxa that seem beneficial to larval health. Such data will ultimately help microbial management in shrimp hatcheries.

To this aim, we have monitored daily the microbial composition of the rearing water, containing larvae raised with or without antibiotics, to investigate if any microbial families were associated with a certain mortality rate at a given larval stage. We also investigated if the antibiotic addition and the natural seawater microbiota influenced the active microbial communities inhabiting the rearing water.

In the context of this study, we identified several specific biomarkers of a given larval stage and health, that might be used as early routine detection proxies in the natural seawater and then during the first days of larval rearing.

## MATERIALS & METHODS

### Study design, water collection and storage

The study was conducted in an experimental shrimp hatchery hosted in a shrimp farming research facility at the Station Aquacole de Saint Vincent (Boulouparis, New-Caledonia). The experiment was carried out in February 2019, where seawater from the water storages, the larval rearing tanks and the control tank were collected during the same larval rearing cycle. Tanks in the hatchery were filled with natural seawater collected from the Saint Vincent Bay. Natural seawater was pumped through a one cm pore size device into a primary reservoir (ResI). Seawater was then filtered through a 10$\mu$m filter prior to storage into a 2m$^3$ storage container implemented with intensive bubbling (ResII). In New Caledonia, the reproduction of *Penaeus stylirostris* is conducted by artificially inseminating mature females, as described by *Pham et al. (2012)*. The day these inseminations were performed, the hatchery tanks were filled with storage seawater in which 5 g.m$^{-3}$ of EDTA (ethylenediaminetetraacetic acid) were added and intensive bubbling was implemented in all the tanks including the control tank that contained no larvae, antibiotics nor food addition. Each tested condition was carried out in triplicates except for the control. One control tank, three larval tanks without antibiotics addition and three larval tanks with erythromycin (antibiotic) addition were considered. In the latter, 2 ppm of erythromycin were added on Day 0 (D0) after EDTA addition and then on days 3, 5, 7 and 9. The larval feeding protocol was as follow: from zoea 1 to zoea 2, microparticles were added five times per day and frozen *Tetraselmis* sp. were given once a day; from zoea 3 to post larvae, microparticles were added twice a day and living *Artemia* sp. nauplii (between 20 to 40 nauplii/larvae/day) twice a day. No water exchange was applied during the first 10 days of the larval rearing.

Natural seawater from the primary reservoir was sampled before the insemination and seawater from the storage container was sampled on the insemination day; while samples from the rearing tanks were collected daily during 9 days, before the first feeding of the day. All the water samples were collected using a 100µm mesh to avoid any larvae on the filter used for the RNAs extractions. For each sample, 1L of water was filtered on 0.2µm sterile membrane filters (S-Pak; Millipore, Burlington, MA, USA). All filters were stored at −80 °C until RNA extractions.

## Daily determination of the zootechnical parameters

Daily larval survival rates were estimated by counting the number of larvae contained in three distinct sub-samples of 100 mL per tank. The larval stages were determined by the observation of 30 larvae per tank using a binocular magnifying glass. This allowed the calculation of the Larval Stage Index (LSI), using the modified equation of Maddox and Manzi (*Maddox & Manzi, 1976*):

$$\text{LSI} = (0 \times Nii + 1 \times Z1 + 2 \times Z2 + 3 \times Z3 + 4 \times M1 + 5 \times M2 + 6 \times M3 + 7XPL)/N$$

where $Nii$ is the number of larvae observed in the nauplius stage, $Z1$ in the zoea 1 stage, $Z2$ in zoea 2, $Z3$ in zoea 3, $M1$ in mysis 1, $M2$ in mysis $M2$, $M3$ mysis $M3$, PL in post larvae 1; and N corresponds to the total number of observed larvae. The Larval Survival Rate (LSR) was defined by averaging 3 direct counts of the living and dead larvae in 1L of rearing water per tank and per day. The LSR was determined as follow LSR: 100*(counted living larvae / initial number of nauplii).

## RNA extractions, reverse transcriptions, sequencing and sequence processing

RNA extractions were performed using the RNAeasy Powerwater kit (Qiagen, Hilden, Germany) following the manufacturer's information. RNA purity and concentration were checked with NanoDrop™ measurements. Then, the total RNAs were reverse transcribed into complementary DNA (cDNAs) using 10 µL of RNA (200 ng/µl), 1 µl of the reverse transcriptase M-MLV at 200 u/µl (PROMEGA), 2 µl random hexamers 50 µM, 4 µl of Buffer 5X, 2 µl of a mix of dNTP a 10 mM each and 1 µl of Rnase/Dnase free water. All the reverse-transcriptions were carried out in a Veriti™ instrument (Applied Biosystems, Foster City, CA, USA), using the program: 10 min at 25 °C, 2 h at 42 °C and 5 min at 85 °C. The cDNAs were stored at −80 °C until shipping to MrDNA (Molecular Research LP, Shallowater, TX, USA) where the PCR, barcode indexing and sequencing of the V4 hypervariable region of the 16S rRNA molecule were conducted using the universal primer combination 515f-806R (*Caporaso et al., 2011*; *Hugerth et al., 2014*). An Illumina HiSeq sequencing was performed using MrDNA protocols with a 2 x 150 bp paired-end run and an average sequencing depth of 20k raw reads per sample. As described in *Giraud et al. (2021)*, the raw data were first demultiplexed using the fastqSplitter available on the MrDNA website (https://www.Mrdnalab.com/mrdnafreesoftware/fastq-splitter.html) (*Giraud et al., 2021*). Then, the reads were treated using the DADA2 (*Callahan et al., 2016*) package available in the Rstudio software. We kept all the reads with a quality score above 30. The sequences were filtered using a maximum excepted error (maxEE) set at 2, a maximum N (maxN) set at 0,

and a truncation based on quality scores (truncQ) set at 2. The sample inference was done by setting the "pool" argument to true. The chimeras were removed using the consensus method. The taxonomy was assigned using the Silva 138 SSU Ref NR99 database (*Quast et al., 2013*). Once the ASV table was obtained, sequences with no affiliation or affiliated to the Eukaryota, Mitochondria or Chloroplasts were removed before further analysis. All the 16S rRNA data are available in the NCBI SRA repository (BioProject ID PRJNA736535, SRP324193 for all the samples: Biosample: SAMN31027695 to SAMN31027756; except ResI sample available in SRR14825806, Biosample: SAMN19659077).

## Downstream microbial analysis

The alpha diversity was estimated with the Chao 1, Shannon and Inverse Simpson (InvSimpson) indexes using the phyloseq package in RStudio (*McMurdie & Holmes, 2013*), while the Good's coverage was estimated using RStudio. Kruskal-Wallis tests were performed on the alpha diversity indexes to highlight potential significative differences between the kinds of water samples and water treatment, using the rstatix package in Rstudio. The rarefaction curves were built using the Vegan package in Rstudio (*Jari Oksanen, 2022*).

Before further microbial analysis, the ASV table was normalized with the Count Per Million (CPM) method using the edgeR package under the RStudio software. The beta diversity was determined by constructing a PCoA based on a Bray-Curtis dissimilarity matrix and the Ward method using phyloseq packages (*McMurdie & Holmes, 2013*) in Rstudio. Clusters were determined by the construction of ellipses using a confidence level for a multivariate t-distribution set at 80%.

Prior to Venn diagram constructions, we established 5 groups of rearing water samples based on the larval stage: nauplii, zoea or mysis; and on the larval survival rate: good or bad. We considered a good larval survival rate when the daily larval counting was above, equal or slightly below the reference (less than 5% in order to mitigate putative counting errors) (the reference is an average of survival rates calculated for each rearing day using data of 10 years of successful data; Ifremer data, D. Pham, comm. pers., 2008–2018). The 5 final defined groups were the nauplii (Nauplii), the zoea with a good survival rate (later named Zoea Good), the zoea with a bad survival rate (Zoea Bad), the mysis with a good survival rate (Mysis Good) and the mysis with a bad survival rate (Mysis Bad). Venn diagrams were then constructed using the Jvenn web application (*Bardou et al., 2014*) (https://jvenn.toulouse.inrae.fr/app/index.html). In order to identify microbial biomarkers, Linear Discriminant Enalysis (LDA) effect size (LEfSe) (*Segata et al., 2011*) were performed with a threshold set at 4 using the microbiomeMarker package (*Cao et al., 2022*) in RStudio.

## RESULTS

### Zootechnical parameters

Contrasting survival rates were observed between the two treatments after D1 (Fig. 1). Larvae reared with antibiotics showed the best survival rates on D9 with more than 70% of surviving larvae, which is similar to the reference value on this rearing day. Inversely, larvae
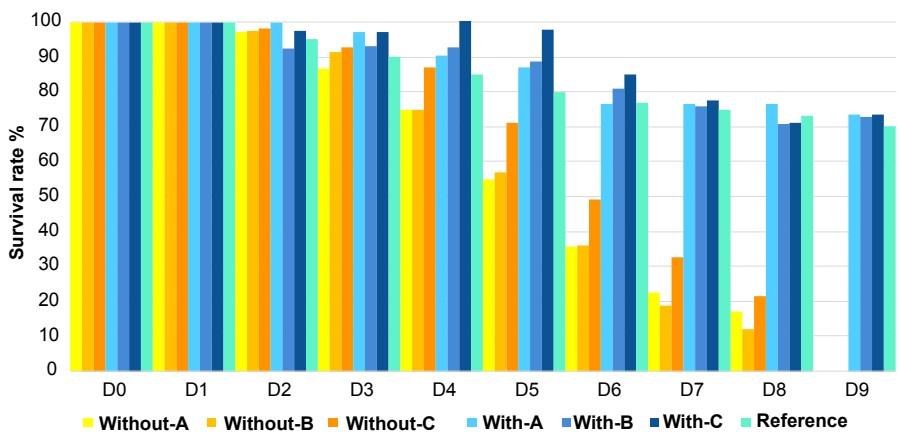

**Figure 1 Evolution of the larval survival during the experiment.** Evolution of the larval survival during the experiment compared to the reference in turquoise (*e.g.*, usual survival rate obtained for a specific day, calculated for each day using data of 10 years of successful rearing; Ifremer data; D. Pham, comm. pers., 2008–2018). Without-A, Without-B, Without-C, correspond to the rearing water without antibiotic in the tanks A, B and C; and With-A, With-B, With-C, stand for the rearing water with antibiotic in the tanks A, B and C. D0 to D9 correspond to the day of the rearing.

raised without antibiotics did not manage to reach the 9th day of rearing as mortalities started to occur on D2 in tanks A and B. In these tanks, larval survival rates only reached 75% on D4 against 85% for the reference. On D5 the larval survival rates in the three replicates without antibiotics were all dramatically below the reference value.

Regarding the larval stage, none of the tanks, with or without antibiotics addition, reached the post larvae stage on D9 but on D10 for all the tanks (except With-C, Table 1). All larvae metamorphosed in mysis on D7, one day later compared to the reference. Except for the tanks B and C reared without antibiotics (Without-B and Without-C in Table 1); all larvae stayed 2 days in the zoea 3 stage (D3 and D4) while it usually only takes 1 day. Globally, compared to the reference, regardless of the rearing condition, larvae had a delay in their metamorphosis. Indeed, at D6 all larvae should have been in Mysis 1 stage while they were still in Zoea 3 stage.

## Time series of the active microbiota in the water

After Eukaryota, Mitochondria, Chloroplasts and unassigned sequences were removed, a total of 21,749,911 reads, spread into 6707 ASVs were obtained from the HiSeq Illumina sequencing of all the samples. The smallest and the largest libraries were respectively composed of 207,221 and 650,143 reads and corresponded to the Control D3 and Control D6 samples (water without larvae, antibiotics nor food).

Overall, the alpha diversity indexes (Table S1) were generally higher in the rearing water without antibiotics than in the rearing water with antibiotics and the control water. Also, the alpha diversity indexes of the control water were higher than the rearing water with antibiotics. The storage water samples (primary reservoir and container storage) were those exhibiting the higher alpha diversity indexes. Despite the rarefaction curves did not reach the plateau for all the samples (Fig. S1), the Good's coverage revealed an overall average

**Table 1 Time series of larval stage.** Time series of larval stage compared to the reference larval stage index to reach each day (e.g., usual larval stage obtained for a specific day; stage reference has been calculated for each day using data of 10 years of successful rearing; Ifremer data, D. Pham, comm. Pers., 2008–2018); D0 to D9 correspond to the day of the rearing, D10 was added to shown that the larvae were mostly at the Post Larvae stage on D10 (excepted in tank With-C). Each color corresponds to a specific larval stage, when in black that means that the larvae were all dead in the tanks. Without-A, Without-B, Without-C, correspond to the rearing tanks without antibiotics; and With-A, With-B, With-C, stand for the rearing tanks with antibiotics. The considering larval stage was named, when more than 75% of observed the larvae were at this given stage.

| | D0 | D1 | D2 | D3 | D4 | D5 | D6 | D7 | D8 | D9 | D10 |
|---|---|---|---|---|---|---|---|---|---|---|---|
| **Without-A** | Nauplii | Nauplii | Zoea 1 | Zoea 2 | Zoea 2 | Zoea 3 | Zoea 3 | Mysis 1 | Mysis 2 | | |
| **Without-B** | Nauplii | Nauplii | Zoea 1 | Zoea 2 | Zoea 2 | Zoea 2 | Zoea 3 | Mysis 1 | Mysis 1 | | |
| **Without-C** | Nauplii | Nauplii | Zoea 1 | Zoea 2 | Zoea 2 | Zoea 2 | Zoea 3 | Mysis 1 | Mysis 2 | | |
| **With-A** | Nauplii | Nauplii | Zoea 1 | Zoea 2 | Zoea 2 | Zoea 3 | Zoea 3 | Mysis 1 | Mysis 1 | Mysis 2 | Post Larvae |
| **With-B** | Nauplii | Nauplii | Zoea 1 | Zoea 2 | Zoea 2 | Zoea 3 | Zoea 3 | Mysis 1 | Mysis 2 | Mysis 3 | Post Larvae |
| **With-C** | Nauplii | Nauplii | Zoea 1 | Zoea 2 | Zoea 2 | Zoea 3 | Zoea 3 | Mysis 1 | Mysis 2 | Mysis 2 | Mysis 3 |
| **Reference** | Nauplii | Nauplii | Zoea 1 | Zoea 2 | Zoea 2 | Zoea 3 | Mysis 1 | Mysis 2 | Mysis 3 | Post Larvae | Post Larvae |

above 99.8% of the total ASV table (Table S1), signifying that the sequencing depth was sufficient.

Kruskal-Wallis tests were performed on the alpha diversity indexes (Table S1) between the different types of water samples: storage water, control, rearing water with antibiotics and rearing water without antibiotic. The tests showed that, for the Chao1 index, the rearing waters with or without antibiotic addition were as significantly different ($p$ value <0.001). For the same indexes, the test exhibited that the control was significantly different from the rearing water without antibiotics ($p$ value = 0.0013); and the storage water was significantly different from the rearing water with antibiotics ($p$ value = 0.005). Considering the Shannon and Inverse Simpson indexes, significant differences were only highlighted between the rearing waters with and without antibiotics, with respective $p$ values <0.001. Samples collected on D1 in the rearing waters with and without antibiotics were compared and the $p$ values equaled 0.513 for the Chao1 indexes. However, significant differences were showed for the Shannon and the Inverse Simpson indexes with $p$ values of 0.049 for both proxies.

Using the whole microbial beta diversity, we visualized how the samples clustered together (Fig. 2). The PCoA displayed 7 clusters (ellipses on Fig. 2), that were built using a confidence level for a multivariate t-distribution set at 80%. The first cluster grouped the control water samples along with the secondary reservoir (ResII). The second group gathered all the rearing water hosting the nauplii. The third cluster encompassed the rearing water samples without antibiotics collected on D2 and D3 and hosting the zoea with a good survival rate; while cluster 4 grouped the rearing water samples without antibiotics collected from D4 to D6 (excepted one sample collected on D4) and hosting the zoea with a bad survival rate. The rearing water samples without antibiotics collected on D7 and D8, and hosting the mysis with a bad survival rate, were all grouped in the cluster 5. The rearing water with antibiotics collected on D2 and hosting the zoea with a good survival rate were apart from the cluster 6 that encompassed all the samples collected in the rearing water with antibiotics and hosting the zoea with a good survival rate (D3 to D6). The last cluster grouped all the rearing water samples with antibiotics that hosted

**Table 2  PERMANOVA using Bray-Curtis distance showing that the variation of the microbial community is explained by all the monitored variables: rearing day, treatment (control, rearing water with antibiotics, rearing water without antibiotics), larval survival rate, larval stage, larval stage and survival rate; and the combine variable rearing day and treatment. Only 8.3% of the variability between the samples is not explained by the listed variables.** Bold values indicate significant correlation ($P < 0.05$).

|  | $R^2$ | P-value |
|---|---|---|
| Rearing day | 36% | **0.0001** |
| Treatment | 25.4% | **0.0001** |
| Larval survival rate | 6.5% | **0.0001** |
| Larval stage | 5.3% | **0.0001** |
| Larval stage and survival rate | 5% | **0.0001** |
| Rearing day and treatment | 13.5% | **0.0001** |
| Residual | 8.3% | |
| Total | 100% | |

the mysis with a good survival rate and corresponded to the samples collected from D7 to D9. The primary reservoir that encompassed the lagoon seawater was aside all clusters (Fig. 2). In order to confirm the PCoA and the clusters, a PERMANOVA along with a pairwise comparison were done to evaluate the water treatment effect on the samples. The PERMANOVA displayed that the sampling day, the treatment (control, antibiotics or not), the larval survival rate, the larval stage, the larval stage and health as well as the day and the treatment together, explained the variability among the samples (Table 2). Only 8.3% of the variability among the samples was not explained by the tested factors. The main factors describing the variability were the sampling day and the water treatment that respectively explained 36% and 25.4% of the variability. The water treatment effect on the microbial diversity, was analyzed with a pairwise comparison using Kruskal-Wallis tests which exhibited that all the treatments: control, antibiotics and not antibiotics, were significantly different (*p*-value <0.001).

The active microbiota of the primary reservoir (ResI), that contained lagoon seawater, was highly different from all the other samples, as most of its lineages belonged to families that were not represented by more than 1% of the total abundance in the other samples (Fig. 3). On D0, the microbial composition of the control water, which contained no larvae, antibiotics nor food, was identical to those of the storage container ResII and of the rearing waters with and without antibiotics collected on D1. The *Alteromonadaceae* and the *Rhodobacteraceae* were dominant in all these samples. Their abundances remained high during the other nine days of rearing with, however, an increased proportion of *Pseudohongiellacaea* and *Bdellovibrionaceae* (Fig. 3). The active microbial compositions of the three replicate samples for each condition displayed homogenous profiles through the whole rearing excepted on D8 for the samples collected in the rearing water without antibiotics (Fig. 3). The active microbiota of the water samples exhibited different compositions and dynamics according to the rearing day, as microbial shifts occurred daily and according to the addition of antibiotics. However, from D2 to the end of the rearing, in the presence or not of antibiotics, the *Rhodobacteraceae* had high abundances (Fig. 3). The active microbial composition of the rearing water without antibiotics encompassed

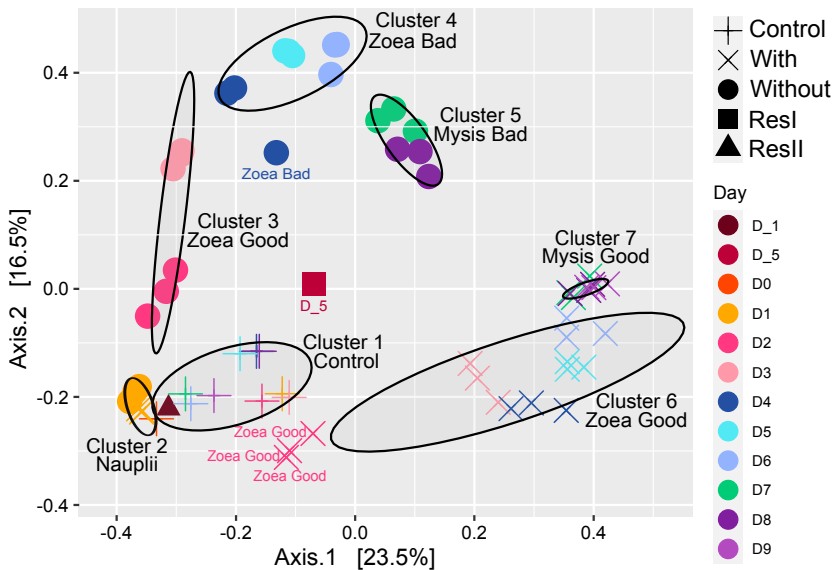

**Figure 2** **Ordination of the water samples based on the PCoA method and a Bray-Curtis dissimilarity matrix.** PCoA of the rearing water samples. The ellipses were constructed using a confidence level for a multivariate t-distribution set at 80%. For each cluster or samples outside the clusters, the larval stage and health (Good for high survival rate and Bad for high mortality rate) are noted. Each color corresponds to a sampling day in the figure and the corresponding sampling day is noted in the same color in the caption on the right side of the PCoA.

mostly members of the *Rhodobacteraceae*, the *Alteromonadaceae*, the *Saprospiracaea* and the *Litoricolaceae* on D1and on D2 (Fig. 3). On D3, a shift occurred with decreasing proportions of *Alteromonacaeae* accompanied with increasing proportions of *Saprospiraceae*. Until D5, the most abundant families were the *Rhodobacteraceae*, the *Alteromonadaceae* and the *Saprospiracaeae*. From D6 to D8, *Alteromonadaceae* and *Saprospiraceae* decreased drastically. On D7 the main families were the *Rhodobacteraceae* and an unknown family related to the ASV19 affiliated to the *Gammaproteobacteria*. On D8, the dominant families varied among the three tanks, where in addition to the *Rhodobacteraceae*, the proportion of *Flavobacteriaceae*, *Vibrionaceae* and/or *Pseudoalteromonadaceae* increased. In the rearing water with antibiotics, on D1, the taxa affiliated to the *Rhodobacteraceae*, the *Alteromonadaceae* and the *Litoricolaceae* were the most abundant; while on D2, the main families were the *Rhodobacteraceae,* the *Alteromonadaceae* and the NS11-12 marine group (Fig. 3). On D3, a shift occurred with an increased proportion of *Pseudoalteromonadaceae* and *Cryomorphaceae*, associated with a drop of the *Alteromonadaceae*. On D4, the *Pseudoalteromonadaceae* decreased whereas the unknown family UBA12409 increased. On D5, the microbiota was dominated by members of the *Rhodobacteraceae* and the *Sapropsiraceae*. On D6, *Rhodobacteraceae* and *Cryomorphaceae* greatly composed the microbiota. A prevalence of members of the *Rhodobacteraceae* and the *Flavobacteraceae* was noticed on D7; while from D8 to D9, the main families of the rearing water microbiota with antibiotics were the *Rhodobacteraceae*, the *Cryomorphaceae* and the *Flavobacteraceae*.

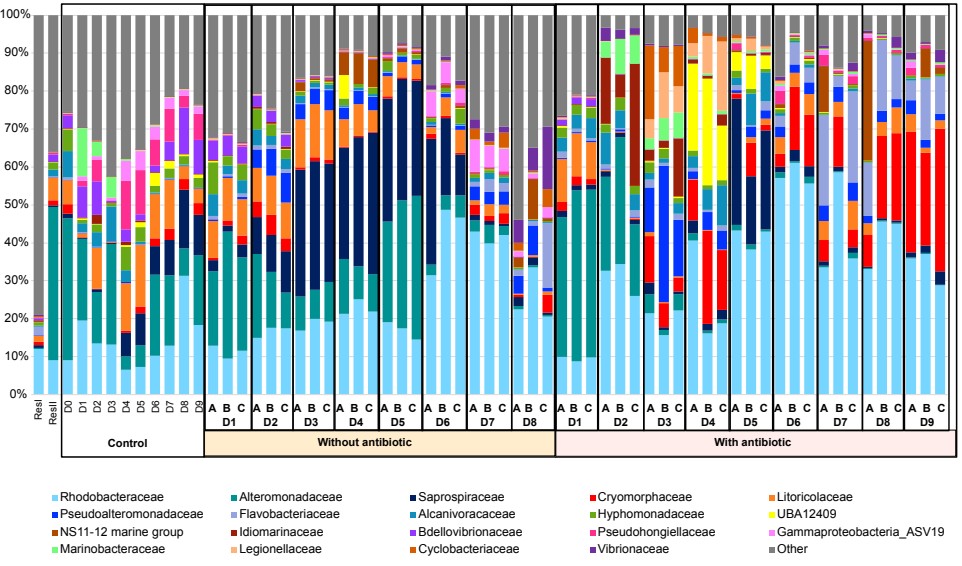

**Figure 3 Microbial composition of the water samples.** Relative abundance of the main prokaryotic families. The relative abundance is represented in terms of percentage of the total prokaryotic sequences per sample. Only families representing more than 1% of the overall abundance in at least three samples are displayed on the barplot. ResI stands for the primary reservoir sample, ResII for the secondary reservoir sample, Control stands for the control water without larvae, antibiotic nor food; without antibiotic for the rearing water without antibiotic, with antibiotic for the rearing water supplemented with antibiotic. D0 to D8 correspond to the sampling day. Sample day are followed by A, B or C which correspond to the replicate tanks for the rearing water with or without antibiotic.

## Specific, shared and core microbiomes among the water samples, larval stages, and health status

In order to determine specific ASVs of a given rearing condition, specific to a larval stage and survival rate, as well as common microbiomes, several Venn diagrams were constructed.

A first Venn diagram was built to compare the rearing water of all larval groups (Fig. 4). As the larval mortalities mainly occurred in the rearing water without antibiotics after D4, the nauplii group was composed by water hosting the nauplii from the 2 rearing conditions. In the same way, the group Zoea Good encompassed all the samples collected in the rearing water with antibiotics from D2 to D6 and samples collected on D2 and on D3 in the rearing water without antibiotics. Thus, the nauplii-specific microbiome contained 658 ASVs. The diagram showed that no ASV was specific to the Zoea Good, while the Zoea Bad gathered 95 ASVs. The Mysis Good condition exhibited 79 common ASVs and the Mysis Bad had 141 specific ASVs. The diagram also displayed a core microbiota composed by 137 ASVs. Specific microbiotas of a given condition as well as a core microbiome were pointed out through the Venn diagram comparisons. The natural lagoon seawater stocked in the primary reservoir and in the storage container, was used for the rearing. These two water storages as well as the evidenced specific microbiotas and the core microbiome were therefore compared. Consequently, the specific microbiota of the rearing water hosting the nauplii had 85 unique ASVs, while 573 ASVs were already present in the natural seawater (Fig. 4B). The comparison between the specific microbiota of the Zoea Bad and the natural

seawater highlighted that 58 ASVs were previously not in the natural seawater whereas 37 ASVs from the natural seawater were only detected in this condition (Fig. 4C). The Mysis Good condition had 48 specific ASVs and shared 31 ASVs with the natural seawater (Fig. 4D). The Mysis Bad condition exhibited 79 specific ASVs and co-owned 62 ASVs with the natural seawater (Fig. 4E). The comparison between the rearing water core microbiome and the natural seawater displayed that only one ASV was specific to the rearing water core microbiome while the 136 other ASVs were shared with the natural seawater (Fig. 4F). Together, these comparisons exhibited the great role of the natural seawater on the rearing water microbiota, as several ASVs detected in the lagoon seawater were also detected at several times of the rearing according to the larval stage and the larval health status. As we observed a microbial partitioning (Fig. 2) with specific microbiotas associated with a given larval stage and survival rate in the rearing water (Fig. 4), we constructed an LEfSe to investigate how the larval stage, the larval health or both interacted with the active rearing water microbiota at the family level.

## Biomarkers at the genus level according to the larval stages and health status

Two LEfSe analysis were conducted to investigate the differentially abundant genus in the rearing water according to the larval stage (Zoea or Mysis) and the larval health status (Good or Bad survival) (Fig. 5A and 5B). A first LEfSe was constructed by analyzing 3 groups: the zoea that were always healthy during the zoea stage (NTA0 Zoea Good in the Fig. 5A), the zoea that were healthy only at the beginning of the zoea stage (NTSA Zoea Good, Fig. 5A); and the zoea that were unhealthy at the end of the zoea stage (NTSA Zoea Bad, Fig. 5A). The second LEfSe compared the rearing water samples hosting the mysis with a good survival rate (Mysis Good) and the rearing water hosting the mysis with a bad survival rate (Mysis Bad). They both permitted to distinguish biomarkers of each condition (Figs. 5A and 5B). Thus, 11 biomarkers were specific of the rearing water hosting zoea good, seven biomarkers of the healthy zoea that became unhealthy, nine biomarkers of the unhealthy zoea, eight of the rearing water hosting mysis with a good survival rate; and 12 of the water hosting mysis with a bad survival rate. Three biomarkers of the zoea that were healthy during the zoea stage (NTA0 Zoea Good) were also statistically enriched in the mysis good condition: *Nautella* and *Lesingera* genera as well as an unknown genus of the *Cryomorphaceae* (ASV12). The genus *Fabibacter* was a biomarker of the healthy zoea (NTA0 Zoea Good), and was later found as a biomarker of the rearing water hosting mysis with a bad survival rate. The HIMB11 group was statistically enriched in the water with the zoea that were only healthy at the beginning of the zoea stage (NTSA Zoea Good), and was also a biomarker of rearing water with unhealthy mysis. Two lineages, *Phaeodactylibacter* and an unknown genus of the *Rhodobacteraceae* (ASV6), were specifically abundant in the rearing water of unhealthy zoea (NTSA Zoea Bad); while they were biomarkers of the rearing water with healthy mysis. Four taxa enriched in the rearing water of the unhealthy zoea at the end of the zoea stage, were also enriched in the rearing water hosting the mysis with a bad survival rate: *Aestuariicoccus* and *Marivita* genera, an unknown genus of the NRL2 (ASV44) and an unknown genus of the *Gammaproteobacteria* (ASV19).

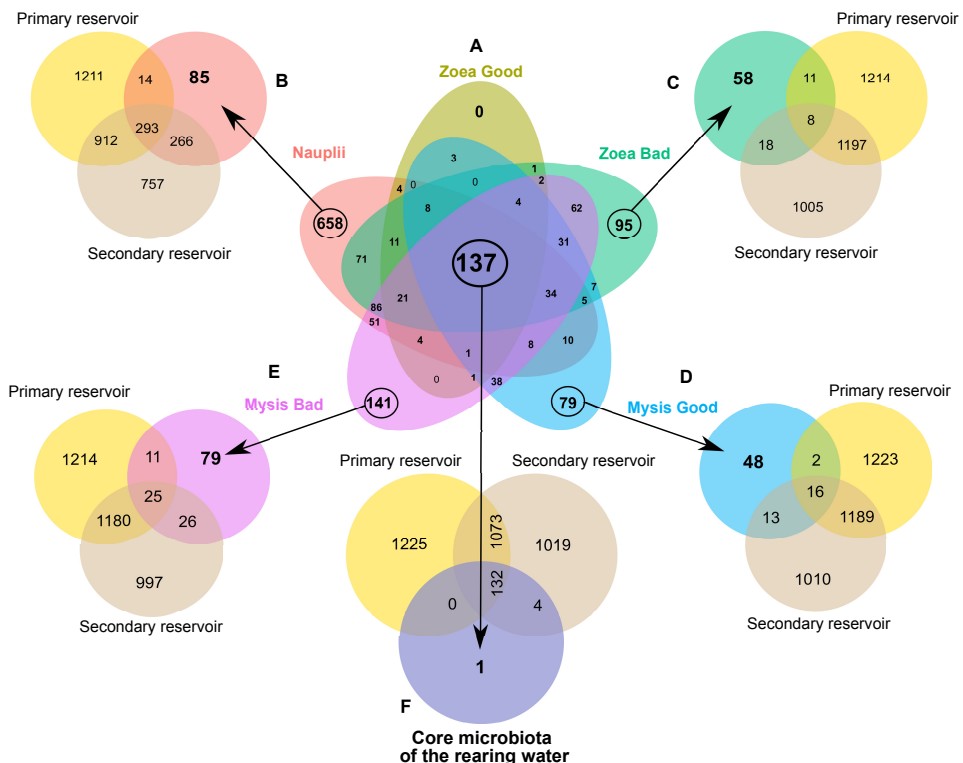

**Figure 4** **Microbial communities associated with the specific and core microbiomes of the whole rearing experiment in the rearing water and the water storages.** (A) Venn diagram of shared ASVs among all the rearing water samples. Coloured ellipses correspond to group-specific ASVs of the rearing water hosting in: red = nauplii, khaki = zoea with a good survival rate, green = zoea with a bad survival rate, blue = mysis with a good survival rate, purple = mysis with a bad survival rate. The overlapping area between all the ellipses, corresponds to the core microbiome composed by the 137 ASVs common to all the samples. The numbers inside the ellipses and in the overlapping zones correspond to the total number of ASVs present in the condition. (B to F) Venn diagram of shared ASVs between the specific ASVs of the water storage (yellow ellipse = group-specific ASVs of the primary reservoir and maroon = group-specific ASVs of the secondary reservoir) and (B) with the nauplii (red ellipse), (C) with the zoea with a bad survival rate (khaki ellipse), (D) with the mysis with a good survival rate (blue ellipse), (E) with the mysis with a bad survival rate (purple ellipse). (F) Venn diagram of the core microbiota of the rearing water and the ASVs of the water storages: in light yellow the primary reservoir and in beige the secondary reservoir.

## DISCUSSION

In this study, we aimed to distinguish, in the rearing water, active biomarkers that were specific of a given larval stage and health condition. In order to assess the metabolically active biomarkers, we extracted the total RNAs from the samples. Using RNA instead of DNA allows to detect recent populations and living assemblages in an ecosystem (*Cristescu, 2019*), as RNA persistence in environment is estimated between 13 to 24 h against months or years for DNA (*Willerslev et al., 2007*; *Marshall, Vanderploeg & Chaganti, 2021*), although ancient RNA has been found in fossils or in sediments (*Orsi, Biddle & Edgcomb, 2013*; *Cristescu, 2019*). Also, several publications using cDNA metabarcoding proved that RNA seemed to be a useful tool to identify living organisms as well as to perform survey

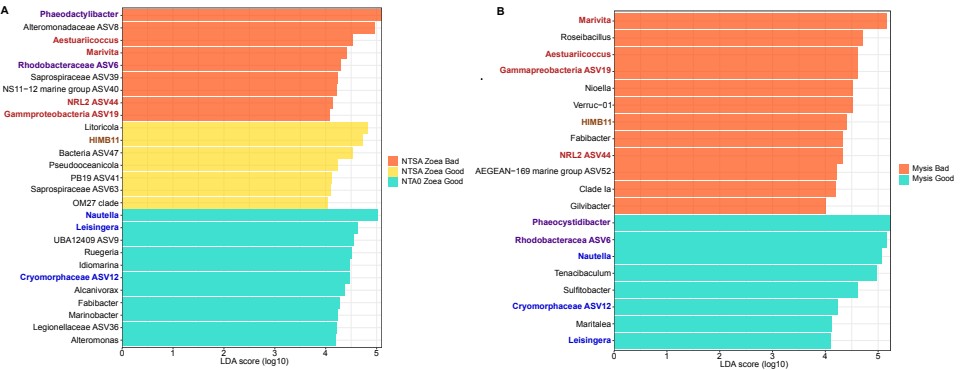

**Figure 5 Differentially abundant genera according to the larval growth and survival status.** (A) LEfSe, linear discriminant analysis (LDA) effect size, exhibiting the genera significantly more abundant in the rearing water A) hosting zoea that stayed healthy during the zoea stage (NTA0 Zoea G g ood), hosting zoea that were healthy at the beginning of the zoea stage and unhealthy at the end (NTSA Zoea Good, corresponding to the rearing day 2 and 3) and hosting zoea with high mortality rate at the end of the zoea stage (NTSA Zoea Bad, corresponding to the rearing day 4 to 6); (B) hosting healthy mysis (Mysis Good) and unhealthy mysis (Mysis Bad). Genera wrote in blue are biomarkers enriched in the rearing water hosting both healthy zoea and mysis. Genera wrote in red are enriched in the rearing water hosting both unhealthy zoea and mysis. Genera wrote in purple were detected as biomarkers of the unhealthy zoea and healthy mysis. Genus wrote in brown was enriched in the rearing water hosting zoea that were healthy at the beginning of the zoea stage and unhealthy at the end and in the rearing water with unhealthy mysis.

and biological monitoring (*Laroche et al., 2018*; *Amarasiri et al., 2021*; *Miyata et al., 2021*; *Miyata et al., 2022*; *Veilleux, Misutka & Glover, 2021*). For these reasons, we performed RNA extractions coupled with reverse-transcription into cDNA, to detect living microbial populations inhabiting the rearing water, as these communities may interact with the composition of the rearing water and with the larvae. In our study, a unique protocol regarding RNA extraction, cDNA retro-transcription, sequencing and sequencing analysis was applied. This is crucial as it allows to perform comparisons and to decrease bias. Indeed, regarding the extraction method used, the quantity, purity and yield of RNA can differ as well as the microbial diversity (*Shu et al., 2014*; *Muto et al., 2017*). The V4 hypervariable region of the 16S rRNA gene has been selected for sequencing because its size, around 254 bp, is quite stable among the prokaryotes and allows HiSeq sequencing (2x150 pb). Also, this region produces optimal procaryote community clustering along with a suitable resolution (*Caporaso et al., 2011*). The choice of the hypervariable region of the 16S rRNA gene is known to influence the microbial taxonomy and the following data interpretation (*Cornejo-Granados et al., 2017*; *García-López et al., 2020*; *O'Dea et al., 2021*). However, *García-López et al. (2020)* showed that when exploring the shrimp microbiota, the use of only a single hypervariable region of the 16S rRNA gene allows a suitable and sufficient description of the microbial community. According to the same authors, data from unique hypervariable sequencing could also allow to monitor microorganism dynamics and to detect potential pathogens along with the development of surveillance tools. To date, 6 studies described the microbial diversity in shrimp larvae (*Zheng et al., 2017*; *Xue et al., 2018*; *Wang et al., 2020a*; *Wang et al., 2020b*; *Giraud et al., 2021*; *Giraud et al., 2022*) and

among them, 5 used the V4 hypervariable region of the 16S rRNA gene (*Xue et al., 2018*; *Wang et al., 2020a*; *Wang et al., 2020b*; *Giraud et al., 2021*; *Giraud et al., 2022*). Besides, 4 studies have dealt with the microbiota of the larval rearing water (*Zheng et al., 2017*; *Wang et al., 2020b*; *Heyse et al., 2021*; *Callac et al., 2022*) and 2 used the V4 hypervariable region of the 16S rRNA gene (*Wang et al., 2020b*; *Callac et al., 2022*). In addition, the V4 hypervariable region of the 16S rRNA gene is often used to describe the microbial diversity in shrimp related studies (for example: *Zheng et al., 2016*; *Hou et al., 2018*; *Xue et al., 2018*; *Wang et al., 2020a*; *Giraud et al., 2021*; *Giraud et al., 2022*; *Callac et al., 2022*). All these studies argue in favor of the use of the V4 hypervariable region: adequate description of the microbial diversity linked with shrimps, suitable for microbiota monitoring along with its wide use in shrimp-related studies.

## Main families of the rearing water microbiota

A noticeable dynamic of the active rearing water microbiota was pointed out either in the presence or not of antibiotics (Figs. 2 and 3), through the rearing. As the PCoA displayed sampling day separated clusters (Fig. 2), it was not surprising that the PERMANOVA indicated that the sampling day influenced up to 36% of the variability among the samples. Despite a clear distinction of the rearing water microbiota between the water with antibiotics harboring heathy larvae and the water without antibiotics presenting unhealthy larvae, several main families were common between the 2 rearing conditions. The presence of main common families in the rearing water where antibiotic or not was added, was similar to what was shown in the rearing water of *Penaeus stylirostris* reared using water filtered on 5 μm and 1 μm and passed through a UV chamber before filling the tanks (*Callac et al., 2022*). The dominant families were the *Rhodobacteraceae* (*Alphaproteobacteria*), the *Alteromonadaceae* (*Gammaproteobacteria*), the *Saprospiraceae* (*Bacteroidia*) and in less extent the *Litoricolaceae* (*Gammaproteobacteria*), the *Cryomorphaceae* (*Bacteroidia*) and the *Flavobacteriaceae* (*Bacteroidia*) (Fig. 3). Overall, during the rearing, the proportion of *Bacteroidia* increased while the part of the *Alpha-* and *Gamma-propteobacteria* decreased, which was similarly highlighted in the rearing water of *Penaeus stylirostris* reared using filtered and UV treated water before filling the tanks (*Callac et al., 2022*); but also, in the rearing water of *Penaeus vannamei* (*Zheng et al., 2017*) and *Penaeus monodon* (*Angthong et al., 2020*). *Rhodobacteraceae* are often detected in the rearing water of marine shrimp larvae such as *P. vannamei* (*Zheng et al., 2017*; *Heyse et al., 2021*), and *P. stylirostris* (*Giraud et al., 2021*; *Giraud et al., 2022*); as well as in the control tank containing only the rearing water used to rear *P. stylirostris* larvae (*Callac et al., 2022*). *Alteromonadaceae* have been previously detected in the rearing water of *P. stylirostris* (*Giraud et al., 2022*) and in the rearing water of *P. vannamei* larvae, until stage Mysis 1 (*Zheng et al., 2017*).

When comparing the microbial diversity of the control tank with the rearing water, through the entire rearing, the microbiota dynamic was different, mostly due to the absence of larvae, food and treatment (PERMANOVA and pairwise comparisons). The same trend was observed in a previous study regarding the rearing water of *P. stylirostris* and its control (*Callac et al., 2022*). The eutrophication of the rearing water with the larvae, contrary to the control water, which stayed oligotrophic during the rearing, could influence the dynamic of
the families presents in all the conditions. Thus, the detection of the *Cryomorphaceae* and their increasing abundance at the end of the rearing, especially in the rearing water with antibiotics (Fig. 3); is in accordance with their features as they are often found in organic rich oceanic water (*Bowman & McMeekin, 2015*). In the same way, that might explain the prevalence of the *Litoricolaceae* in the control water and in the first days of the larval rearing, as members of this family are known to grow on oligotrophic medium (*Webb et al., 2014*). Another family affiliated to the *Flavobacteriaceae*, appeared in the rearing water with larvae and became among the most abundant at the end of the experiment in the rearing water with antibiotics hosting the mysis with a good survival rate. This trend differed from the other study done in the rearing water of *P. stylirostris* larvae in which important larval mortalities were observed in the rearing water with antibiotics or without antibiotics (*Callac et al., 2022*). That also differs from the study made by (*Zheng et al., 2017*) where the *Flavobacteriaceae* were present in high abundance in the rearing water of *P. vannamei* larvae at the zoea stage (*Zheng et al., 2017*). Thus, the dynamics of the bacterial families in the rearing water appeared to be related to the sampling day, the rearing water quality and to the addition of antibiotics.

## Antibiotics and larval health as drivers of the rearing water microbiota?

Despite the dominance of several main families, the PCoA (Fig. 2) and the statistical analysis (PERMANOVA and pairwise comparison) exhibited that the treatment (control *versus* rearing water with antibiotics *versus* rearing water without antibiotics) influenced the active microbial diversity of the rearing water and influenced up to 25.4% of the variability among the samples while the survival rate accounted only for 6.5%. Even if the statistical analysis shows a bigger effect of the treatment on the active microbial diversity, it is hard to untangle the effects of the antibiotics addition and of the larval death rate on the rearing water microbiota. Indeed, it is complex to discriminate which community was impacted by the treatment rather than by the larval survival. However, it seemed that the antibiotic addition highly influenced the larval survival rate (Figs. 2 and 3). Antibiotics use is a worldwide habit in shrimp hatcheries as they are used either to avoid larval mortalities caused by pathogenic *Vibrio* species (*Holmström et al., 2003*; *Aftabuddin et al., 2009*) or for prophylactic reasons under veterinary instructions. The effect of the antibiotics on the microbiota of the rearing water of aquacultured animal is poorly documented, while several studies have investigated its effect on animal heath, physiology or microbiota (*Kim et al., 2019*; *Zeng et al., 2019*; *Holt et al., 2021*; *Yukgehnaish et al., 2020*). To this day, only one study has dealt with antibiotic effects on the rearing water microbiota of the Penaeids (*Callac et al., 2022*). The authors showed that the antibiotic addition had a significant effect on the microbial diversity of the rearing water on D1 before the larval mortalities occurred. Antibiotic addition at zoea stage during the rearing of the freshwater shrimp *Macrobrachium rosenbergii,* also induced discrepancy among the abundance of the main microbial genera in comparison with a rearing without antibiotics (*Ma et al., 2020*).

Besides the influence of the antibiotic addition on the active microbial diversity of the rearing water, larval mortalities might also affect the composition of the rearing water microbiota. This might especially happen, like in our study, when massive mortalities

occur (Fig. 1) and no dead larvae removal or water renewal are applied. Previous studies have proved that animal death (or death of any living organism) implied changes of their microbiota (*Preiswerk, Walser & Ebert, 2018*; *Benbow et al., 2019*). Indeed, the microbiome associated with the living host changes after death and let's place to the necrobiome, which influences its closest environment, by decomposing the host-derived organic matter (*Cobaugh, Schaeffer & De Bruyn, 2015*; *Benbow et al., 2019*). Therefore, in turn, the microbiota of the closest surrounding of the dead organisms also changes (*Cobaugh, Schaeffer & De Bruyn, 2015*; *Finley et al., 2016*; *Lobb et al., 2020*). In our case, in the rearing water exhibiting high mortality, we can assume that the decaying of the dead larvae can modify the rearing water composition, as well as the water microbiota with the release of the necrobiome in the rearing water. Such process could have started since D2 with the beginning of the mortalities, and continued until D9 when no living larvae remained in the tanks (tanks without antibiotic, Table 1). We can also hypothesize that among the biomarkers or ASVs specific of the unhealthy conditions, some of them were related to the necrobiome. In our study, the use of antibiotics seemed to prevent the larval mortality (Fig. 1), and also appeared to influence the rearing water microbiota (Table 2) along with the larval survival rate, larval stage and possibly the necrobiome.

## Interactions between the natural seawater, the rearing water and the larval stage and health

In the light of our data, we established that various taxa of the rearing water were specific to a larval stage and of a larval health; except for the zoea with a good survival rate for which no specific ASVs were found (Fig. 4). This also exhibited a microbiota partitioning, as also shown in the rearing water of *P. stylirostris* larvae (*Callac et al., 2022*) and *P. vannamei* larvae (*Zheng et al., 2017*), revealing the great importance of the larval stage and of the larval health on the rearing water microbiota. In addition, *Giraud et al. (2021)* and *Giraud et al. (2022)* have shown that a horizontal transmission occurs between the shrimp larvae and their surrounding rearing water, suggesting a putative dynamic between the larval microbiota and the rearing water microbiota.

According to the specific microbiota of a given larval stage and survival rate, many of the specific evidenced ASVs were previously detected in the natural seawater and in the reservoir (Fig. 4), suggesting the high importance of the lagoon and of the storage waters on the rearing water microbiota, as already shown in *Callac et al. (2022)*. In addition, several studies have showed that multiple lineages were shared between the shrimp early life stages and the water reservoirs (*Zheng et al., 2017*; *Wang et al., 2020b*; *Giraud et al., 2021*; *Giraud et al., 2022*), suggesting that microbial exchanges occurred between the rearing water and the larvae (*Giraud et al., 2021*; *Giraud et al., 2022*). Such exchanges and interactions, might also occur with the necrobiome in the rearing tanks which exhibited unusual larval death. The rearing water is thus a complex ecosystem where various interactions take place among the water and also between the water and the larvae according to their stage and health status. Seeking for significant and reliable microbial biomarkers might help to monitor and to predict the fate of upcoming rearing.

## Proxy uncovering: specific biomarker identification of a given rearing condition

Our main objective was to unveil biomarkers at the genus level specific to a larval stage and health status, to later use them to monitor larval health and manage the rearing water.

In our study, we have highlighted that the use of antibiotics seemed to prevent the larval mortality (Fig. 1), and to influence the rearing water microbiota (Table 2) along with the larval survival rate and the larval stage. Together, by comparing the active microbiota of the rearing water hosting zoea or mysis with various survival rates with LEfSe analysis, we unveil specific biomarkers of a given larval stage and health (Fig. 5).

The *Nautella* and the *Leisingera* genera were both detected as biomarkers of the healthy zoea and mysis (Fig. 5). Interestingly, the genus *Nautella* has been identified as biomarker of diseased *P. vannamei* larvae and their rearing water (*Zheng et al., 2016*; *Zheng et al., 2017*); while other studies rather exhibited this genus as a biomarker of healthy larvae and shrimps (*Wang et al., 2020a*; *Restrepo et al., 2021*). The high abundance of this controversial genus in the rearing water seemed to be in our case a biomarker of healthy larvae. The genus *Leisingera*, biomarker of the rearing water with healthy zoea and mysis larvae, is known to produce secondary metabolites with antimicrobial activity against several *Vibrio* species (*Gromek et al., 2016*) and has been detected in the gut of the shrimp *P. vannamei*, in the eggs of the hawaiian bobtail squid and in Pacific oyster larvae (*Gromek et al., 2016*; *Duan et al., 2021*; *Fallet et al., 2022*). Beside *Lesingera*, others genera, such as *Ruegerira*, *Alconivorax* and *Marinobacter* might have a probiotic activity in the rearing water hosting the healthy zoea. Indeed, *Ruegira* exhibited antagonist effects against fish vibrio such as *Vibrio anguillarum* and other bacteria isolated in a fish farm (*Porsby, Nielsen & Gram, 2008*; *Sonnenschein et al., 2017*). It has been shown that the larvae of the shrimp *P. indicus* had a better growth, metamorphosis rate and survival rate when fed with microalgae associated with bacteria affiliated to *Alteromonas* and *Marinobacter* genera (*Sandhya, Sandeep & Vijayan, 2020*). The addition of *Alteromonas macleolii* 0444 during the rearing of oyster and scallop larvae showed a protection of the larvae during *Vibrio* challenges (*Kesarcodi-Watson et al., 2012*). *Tenacibaculum* was significantly enriched in the rearing water hosting the healthy mysis (Fig. 5). In a similar way as the *Nautella,* the genus *Tenacibaculum* has been shown to be a biomarker of diseased *P. vannamei* affected by the "cotton shrimp-like" disease (*Zhou et al., 2019*). However, *Tenacibaculum* was amongst the six most abundant genera detected in both the healthy *P. vannamei* larvae and their rearing water (*Zheng et al., 2016*). As for the *Nautella,* the high prevalence of *Tenacibaculum* in the rearing water with healthy mysis suggests them to be a biomarker of healthy mysis. We can hypothesis the same about the unknown genus related to the ASV12 and affiliated to the *Cryomorphaceae* for which no metabolic or ecologic function can be inferred except a potential beneficial role for larval survival. Together, these taxa: *Nautella Leisingera,* unknown genera related to ASV12, *Ruegerira*, *Alconivorax*, *Marinobacter* and *Tenacibaculum* seemed to be beneficial for the larval survival and maybe for their physiology (enhance immunity, metamorphosis); and might, in the rearing water, outcompete the r-strategist microorganisms and/or putative opportunistic pathogens.

Several biomarkers were detected in the rearing water of unhealthy larvae, and some were present in both the rearing water with unhealthy zoea and unhealthy mysis. This is the case for *Marivita* and *Aestuariicocccus* (Fig. 5). While the last genus was so far never found in shrimp rearing water or shrimp microbiota, the *Marivita* were found in high abundance in ponds where *Penaeus vannamei* adults and larvae were reared (*Lin et al., 2017*; *Yang et al., 2018*; *Wang et al., 2020a*). Even if the ecological role of *Marivita* in aquaculture ecosystem remains unknown (*Lin et al., 2017*), it seemed that the presence of these bacteria was related to larvae mortalities in our study. The same conclusion can be made for the unknown genus related to the ASV19 (*Gammaproteobacteria*) and for the unknown genus related to the ASV44 (NRL2), distinguished as biomarkers in the rearing water hosting the zoea and the mysis with bad survival rate (Fig. 5). The HIMB11 was a biomarker of the healthy zoea that later became unhealthy and of the mysis with a bad survival rate (Fig. 5); this biomarker might be related to upcoming and occurring larval mortalities. *Nioella*, one of the biomarkers of the mysis with a low survival rate in the rearing water, is a genus that has been detected in the gut microbiota of *P. vannamei* affected by the white feces syndrome (*Lu et al., 2020*). This disease is due to a polymicrobial pathogens infection and *Nioella* seemed to be in diseased-specific associations with species related to *Vibrio tubiashii* and *V. coralliilyticus* (*Lu et al., 2020*). Interestingly, the *Fabibacter* genus was enriched in rearing water with healthy zoea and in the rearing water hosting mysis with a bad survival rate (Fig. 5). Similarly, the *Phaeocystidibacter* and an unknown genus related to the ASV6 (*Rhodobacteraceae*), were found as biomarker of the zoea with a bad survival and of the healthy mysis in the rearing water (Fig. 5). These contrasted behaviors suggest that either these biomarkers had to be used only at these specific moments of the rearing (with the specific larval health and stage of the detected biomarkers: *Phaeocystidibacter* and an unknown genus related to the ASV6) or are not reliable biomarkers of larval health.

One can argue that biomarkers might also reflect the observed one day larval metamorphose occurring during mysis and post-larvae transformations (D6 and D9). However, it is quite common to observe a transition phase from zoea to mysis on Day 6; and from mysis to post-larvae on Day 9 (reference Table 1). As our larval observations took place in the morning, we probably missed the metamorphosis peak occurring at D6 and D9. Indeed, as shown in the Table 1, all the larvae reached the mysis 1 stage at D7 and all the larvae from the tanks reared with antibiotics reached the post larvae stage on Day 10 (except tank C which was still transitioning mysis 3, post larvae) (Table 1). Therefore, we can argue that the detected biomarkers were specific of a given larval stage and survival rate.

At the writing time, larval mortalities are still occurring in the territory's hatcheries highlighting the great importance of defining reliable proxies that can be used as early surveillance of the rearing water or prior to the rearing by monitoring the lagoon seawater. Thus, these biomarkers aim to help microbial management of the rearing waters by suggesting new probiotic populations or beneficial taxa to improve water quality or larval health.

Further studies in metatranscriptomic should be done on the rearing water microbiota and on both larvae and their microbiota, in order to highlight the genes that are specifically

enriched according to each rearing condition. Such data will allow to investigate which pathways are differentially expressed according to the larval stage and health when contrasted survival rate are observed.

## CONCLUSIONS

Our findings exhibited that shrimp larval rearing water is a complex and dynamic ecosystem, driven by several parameters: the original natural seawater, the presence or not of antibiotics, the larval stage, the larval health status and maybe by the necrobiome. We also highlighted that it is hard to untangle the effects of the antibiotic addition and of the larval mortalities on the rearing water microbiota, especially in the case of mass mortalities occurring in the rearing water without antibiotics in comparison with great survivals in the rearing condition with antibiotics. In addition, our results revealed that, given a larval stage and survival rate, several active taxa were specific to these considered parameters (except for the zoea with a good survival rate). Among these lineages, many of them were originally detected in the natural seawater. That outcome disclosed the great importance of the natural seawater microbiota on the rearing water microbiota. We also showed that the necrobiome associated with dead larvae might potentially impact the structure of the rearing water microbiota. The biomarker investigation allowed to highlight that several genera: *Nautella, Leisingera,* unknown genera related to ASV12 (*Cryomorphaceae*), *Ruegerira, Alconivorax, Marinobacter* and *Tenacibaculum*, could potentially be beneficial for the larval survival and physiology; and may, in the rearing water, overcome the r-strategist microorganisms and/or putative opportunistic pathogens. Members of these genera might also act as probiotics for the larvae. On the contrary, *Marivita, Aestuariicocccus,* an unknown genus related to the ASV19 (*Gammaproteobacteria*), an unknown genus related to the ASV44 (NRL2), HIMB11 and *Nioella*, appeared to be unfavorable for the larval survival and could be associated with upcoming and occurring larval mortalities. To further understand the role of these specific genera in the rearing water or on the larvae, several studies such as metatranscriptomic analysis are needed, in particular to uncover their activities and ecological role. Other analysis might be done on the detected beneficial biomarkers to test their putative probiotic activities. Despite the unknown role of these specific genera during the rearing, these biomarkers could be used to design specific qPCR primers and thus, be routine proxies to forecast the larval health. They could be used at the beginning of the rearing and even before, in the natural seawater, as an early warning investigation. Ultimately, the same proxies could also help to monitor and to estimate the evolution of the larval rearing; and to further manage the rearing water microbiota and select beneficial microorganisms for the larvae.

## ACKNOWLEDGEMENTS

We sincerely acknowledge all the members of the Station Aquacole de Saint Vincent for their great help before and during the experiment; and for the zootechnical, technical and analytical support. Special thanks to Maxime Beauvais for his indubitably help for the sampling and sample processing. We particularly thank Jean-René Maillez,

Jean-Sébastien Lam and Julien Le Rohellec for their precious help in the hatchery. We deeply thank Dominique Ansquer, Valentine Ballan and Gwenola Plougoulen for their valuable biomolecular work. We also greatly acknowledge Valérie Perez for her valuable bioinformatic help.

### Funding

This work was supported by the RESSAC project (LEAD-NC, Ifremer New-Caledonia) within the framework agreement with the New Caledonian Provinces and Government and by the Pacific Doctoral School. The funders had no role in study design, data collection and analysis, decision to publish, or preparation of the manuscript. The funders had no role in study design, data collection and analysis, decision to publish, or preparation of the manuscript.

### Grant Disclosures

The following grant information was disclosed by the authors:
The RESSAC project (LEAD-NC, Ifremer New-Caledonia) within the framework agreement with the New Caledonian Provinces and Government and by the Pacific Doctoral School.

### Competing Interests

The authors declare there are no competing interests.

### Author Contributions

- Nolwenn Callac conceived and designed the experiments, performed the experiments, analyzed the data, prepared figures and/or tables, authored or reviewed drafts of the article, and approved the final draft.
- Carolane Giraud analyzed the data, authored or reviewed drafts of the article, and approved the final draft.
- Viviane Boulo, Nelly Wabete and Dominique Pham conceived and designed the experiments, performed the experiments, authored or reviewed drafts of the article, and approved the final draft.

### Data Availability

  The sequences are available at NCBI BioProject: PRJNA736535.

### Supplemental Information

Supplemental information for this article can be found online at http://dx.doi.org/10.7717/peerj.15201#supplemental-information.

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
