# Peer review of "Microbial biomarker detection in shrimp larvae rearing water as putative bio-surveillance proxies in shrimp aquaculture"

_PeerJ, doi:10.7717/peerj.15201_

## Round 0.1 · original submission · Major Revisions

While the science is robust in this manuscript, after review, I have deemed that the English is not of a standard that I can send this out for review. I am therefore requesting that you revise this manuscript and resubmit it.

It is my strong recommendation that you recruit a Scientific Editing service to assist you with this. If you can provide evidence of this when you resubmit with would speed up the review process.

We look forward to reconsidering your manuscript in the near future. Thank you for choosing to submit to this special issue and to PeerJ.

---

## Round 0.2 · Major Revisions

Please carefully address the reviewers concerns regarding inclusion of detailed methods and justification for your selections.

Reviewer 1 ·

Basic reporting

In this manuscript, Callac et al. indicate potential culture-environment biomarkers corresponding to health and survival of Pacific blue shrimp Penaeus stylirostris. Although there are other published studies attempting the same, nearly all of the literature is focused on P. mondon or Litopenaeus vannamei, therefore this work could offer valuable information to a neglected aquaculture sector.

Unfortunately, the current manuscript is a little overwhelming in terms of the bacterial taxa, diversity metrics and pairwise comparisons. Attempting very granular descriptions for each and every sample group (as opposed to identifying key trends and changes) results in a manuscript that is too long. I commend the authors for being so thorough but much of the text and information is unnecessary given what is presented in figures and tables.
In order to make the manuscript more manageable for the reader, you could consider:
- Removing either Chao1 or ACE descriptions from the text (L216-235) – including four alpha diversity measures is a little redundant given their impact on the results of the study.
- Removing the hierarchical clustering analysis (text and figure). This can be incorporated into the PCoA, given they are both based on the same distance. See later comments on how to improve PCoA.
- Rewriting taxonomic proportion shifts on each day as general trends. The same can be done for the section describing Pearson correlations.

I gather the manuscript has been reviewed by a native English speaker; however, the current use of language and grammar is sometimes prohibitive to the reader’s understanding of the text. I have listed a number of minor adjustments in the 'Additional comments' section but these are not exhaustive and the text should be further assessed prior to resubmission. One common phrase is “presence or not of antibiotic” (or some version of this) and often does not fit with the corresponding sentence.

Figures
(Figure 2 and 3 covered above)
Figure 4: It would be useful if the days were more distinguishable. Maybe increase the spacing between groups or include a coloured label for each group.
Figure 5: I really like this figure! But the legend could be more concise. E.g. “Coloured ellipses correspond to group-specific ASVs: red = nauplii, khaki = …”

Experimental design

The manuscript is extensive in its sampling but gives no consideration to the shrimp microbiome itself. Microbes in the water are likely interacting with microbes in the host, potentially leading to increased/decreased chances of survival. And microbiome variation within a community could result in differential responses to environmental biomarkers.
I still believe this is a potentially useful dataset (without the shrimp microbiomes of the corresponding cultivar) however the interplay between the two communities should be discussed.

Was EDTA and bubbling added to the control tank?

Although RT-PCR of RNA to make a cDNA template for amplicon PCR is a perfectly valid approach, it is not the norm for microbiome studies. The authors allude to “active microbiota” but should also take the time to explain the benefits/drawbacks of this approach.

Validity of the findings

Why was metamorphosis delayed? And does this make the biomarkers identified perhaps specific to an atypical culture?

Please be consistent with number of decimal places for p -values. Consider using < 0.001 etc.

It would be much more insightful if the PCoA included samples both with and without antibiotics. That way you could also test clustering with ordination statistics like PERMANOVA/ANOSIM and betadisper. Furthermore, why aren’t the controls (without antibiotics or shrimp) or D0 in the ordination? That would be a useful comparison.

I’m not convinced correlating LEfSe with sample groups is the best approach here. You are ultimately interested in biomarkers of good and bad survival in each sample group (larval stage) and a global LEfSe to look for markers in all samples might miss potential signals. I would suggest using a combination of tools like DESeq2, ANCOMBC and ALDEx and then creating a list of markers from the consensus.

Are families useful for biomarkers? There is sometimes a lot of variation within bacterial families. Can you be sure all genera/species will impact mortality in the same way?

Additional comments

Abstract
“may allow to establish microbial proxies” - may allow the establishment of microbial proxies
“daily monitored the composition…” - “monitored the daily composition …”
“As antibiotics were added or not to the rearing water, two distinct rearing conditions were analyzed.” needs rephrasing.
“considered rearing cycles” is unclear
“highlighte d” – remove space
“specific of a given larval stage” – “specific to a given larval stage”. This needs correcting throughout.
It is unclear of the relationship between storage water and primary reservoir containing lagoon seawater in the abstract. It is later clarified but is confusing when reading the abstract alone.
“they could allow to monitor” – allow who?

Introduction
L59: “… understood yet …” should be “yet understood”. Remove “the” in “trigger the larval death”.
L61: “port larvae” should be “post larvae”
L61-62: include correct citation for data URL
L63+: Does “Territory” need to be capitalised?
L66-67: “wrecked” not appropriate here. Maybe “disproven”.
L68-70: “Furthermore, no larval septicemia was noticed through the various analysis conducted by the Neo-Caledonian network of Shrimp Epidemiological Vigilance (REC-DAVAR)” – no context for this sentence.
L86: “link” should be “links”
L89: remove “-“ from “-oceanic”
L95: “In this aim” should be ‘to this aim”
L95: correct position of “daily” as above
L96: “antibiotic” should be “antibiotics”. Please correct throughout the manuscript where needed.
L97: “to a certain…” should be “with a certain…”
L98: “had a role on the…” should “influenced the…” or “impacted the…”

Materials and Methods:
L114: remove “done’
L116-117: “added and an intensive…” should be “added and intensive…”
L118: “In other words” is unnecessary
L126: remove hyphen in “insemination-and”
L127: remove period in “.;”
L128: “collected daily”
L134: “allowed to calculate the …’ should be “allowed the calculation of the …”
L156-157: The Illumina HiSeq cannot use 300bp reads. Was a MiSeq used?
L160:163: The DADA2 pipeline as a number of steps. Please include more detail, including which pooling method was implemented during sample inference.
L164-165: “eukaryotes/Eukaryota, mitochondria …”
L166-167: Thank you for uploading your sequencing data.
L180-181: why include survival rates if they are “slightly below the reference”?
L189: is “(Cao)” an incomplete reference?

Results
L195: “Contrasted” should be “Contrasting”
L195: contrasting after D1
L195-196: “Larvae reared with antibiotic showed the best survival rates on D9 with more than 70% of surviving larvae” is not true as it is currently written. D0-D7 all have higher survival rates.
L198: It appears from Fig1 that mortalities occurred from D2?
L211: correct “eukaryotas, mitochondrias…” as above
L222-230: ACE could be removed
L236: Clarify by adding “(beta)” diversity
L236: replace “gathered” with “clustered”.
L266: should be “total abundance”
L275-276: should be “microbial shifts occurred daily and according to the addition of antibiotics”
L289: should be “most abundant”.
L300: should be “water samples, larval stages, and health status”
L302: remove ‘per say”
L305: could replace with “… to compare the rearing water of all shrimp groups’ or the equivalent.
L308-309: should be “nauplii group composed of … ”
L315: rephrase “These comparisons allowed to point …”
L322: remove “not”
L329-330: “as several ASVs seemed to be able to re-activate during the rearing according to the larval stage and the larval health status” is an odd sentence. What does re-activate mean? Surely they are just residual taxa from the reservoir and not “activating” depending on shrimp condition?
L331: rephrase “microbial clusetrization”
L337: “A LEFse …”
L340: Remove “were”
L353,L359: “greatly positively” not correct
L361: “in less exten to” should be “to a lesser extent to ..”
L382: Probably best not to use the word “evolution” here
L390: replace “the part of”
L420: “dynamics”
430: remove “or”
L440: rephrase
L514-516: This sentence is unclear and needs rephrasing. The correlogram shows several families with strong correlations with good/bad survival.
Need more comparisons to control – no shrimp.
L524-535
L533: change to “to a lesser extent”
L556: change “allowed to spot”
L569: “in one way or another”
L572: change “permitted to uncover’ to “uncovers’
L584: change “mortalities one on the …” to ‘mortalities on the …”
L591-592: We also showed that the necrobiome associated with dead larvae could also impact the structure of the rearing water microbiota” I don’t think you’ve shown this. It is certainly a valid hypothesis, but you offer no direct measurements.

Reviewer 2 ·

Basic reporting

The paper from Callac et al. aims to characterize the active microbiota associated with rearing water with and without antibiotics using the sequencing of the V4 hypervariable region of the 16S rRNA gene. So far, only a few articles describe the active microbiota in the shrimp microbiota. In line with that, the paper is of interest.

There are a few significant issues with data analysis that prevent supporting the claims and conclusions made in the manuscript.

1. The alpha diversity analysis showed a difference between sample groups. Nevertheless, the author used the total sequencing depth for this analysis. These metrics should be made at the same sequence deep for all the samples because the alpha diversity metrics are strongly influenced by sequence depth. Thus, the authors should clarify that alpha diversity metrics were calculated at the same sequence depth and using at least 1000 iterations to rarify the sequences. In this manner, they can compare the metrics at the same sequence depth. In addition, the authors should include the rarefaction plots for all the libraries at the higher sequence depth as supplementary images. Does the rarefaction analysis reach a plateau?

2. Why the authors selected the V4 16S hypervariable region to describe the microbiota? Please justify.

3. More clear description of the experimental methods and data analysis is necessary. Also, a more extensive comparison of the results and discussion with other findings reported in the literature, particularly concerning using the V4 sequencing from RNA instead of DNA.

Experimental design

Please see section number 1.

Validity of the findings

Please see section number 1.

Reviewer 3 ·

Basic reporting

The manuscript by Callac et al. monitored the composition of the active microbiota of the rearing water in a hatchery of Penaeus stylirostris to distinguis the microbial taxa related to high mortality rates at a given larval time.
The manuscript is well written, consice and with an adequate experimental design to achieve the objectives. I suggest a few minor changes to improve clarity before publication.

Experimental design

The study has a ell defined question and an adequate experimental design to explore it. I suggest to add a few details (aditional comments) to the methods and discussion sections for clarity.

Validity of the findings

The conclusions are consistent and according to the experimental design and analyses. I suggest to add a few details (aditional comments) to the methods and discussion sections for clarity.

Additional comments

In the methods section:
- Seams evident the authors extracted the RNA using the complete larvae during the process. However, please add this detail for clarity.
- Did the authors used a DNAse during the extraction process of the RNA. Please add this information with the necesary details for replication.
- Did the authors used any method to assess the quality of the RNA extracted from the samples? Please add this information with the necesary details.
- During the bioinformatics analysis, what was the minimum read quality used? Please add this information.
In the discussion section:
- According to the Chao1 index, what was the percentage of the recovered bacterial communities, that is, was the sequencing effort sufficient to characterize most of the bacterial communities present in the samples?
- Add a few lines stating the limitations of the study, such as, how relevant are these results when comparint to actual aquaculture conditions?

---

## Round 0.3 · Minor Revisions

Many thanks for your revision. One of the reviewers still has several concerns - can you please address these?

Reviewer 2 ·

Basic reporting

- The authors should again include their statistical analysis on alpha diversity. These results are relevant because the PERMANOVA analysis was based on beta diversity, which is different from the alpha diversity results.

- The authors should discuss further the selection of the V4 region but directly consider literature related to shrimp microbiota. For example, there is a work on hypervariable regions in the shrimp microbiota that discusses the use of different hypervariable region and their impact on shrimp microbiota:
https://pubmed.ncbi.nlm.nih.gov/31963525/

Experimental design

NA

Validity of the findings

NA

Additional comments

NA

Reviewer 3 ·

Basic reporting

The authors have answered the questions adequately. Th manuscript is clear and ready for publication.

Experimental design

The authors have added sufficient details to the manuscript to support the experimental design

Validity of the findings

The authors have added clear and detalied results to support their conclusions.

---

## Round 0.4 · accepted · Accept

Thank you for your attention to detail and your responses. I am pleased that the manuscript is now acceptable for publication. There are some English issues that may need further attention as it moves through the proofing process.